# The Women's Wellness with Type 2 Diabetes Programme: Feasibility of an online peer support and goal-setting intervention for midlife women

Deniz Bozkurt[1]*, Maria Duaso[1], Iliatha Papachristou Nadal[1], Rosie Walker[2], Jackie Sturt[1]

1 Florence Nightingale Faculty of Nursing, Midwifery and Palliative Care, King's College London, London, United Kingdom, 2 Successful Diabetes, Ipswich, United Kingdom

☯ These authors contributed equally to this work.
* denizbaser88@gmail.com

## Abstract

### Objective

The Women's Wellness with Type 2 Diabetes Programme (WWDP) is a complex intervention, encouraging positive lifestyle behaviours to enhance wellness outcomes. A previous feasibility study found the original WWDP had a good effect signal and was acceptable to women but had challenges with national scalability. In response, the programme was revised (WWDP+). This study evaluated the feasibility of delivering the WWDP+ and its evaluation procedures and assessed physical and psychosocial outcomes in midlife women to inform the design of a future randomised controlled trial.

### Methods

A single arm, pre- and post-intervention design were conducted. Women living with type 2 diabetes and aged between 45–65 years were recruited via a social media campaign using purposive sampling targeting individuals who had previously expressed interest in type 2 diabetes research. Alongside recruitment and completion rates, clinical and behavioural study outcomes were assessed at baseline, 3 months and 6 months post baseline. Individualised goals were set at week 3 and assessed at 12-week. Analysis comprised descriptive statistics, Wilcoxon's signed-rank and paired t tests.

### Results

Thirty-four women (mean age = 55.4) from diverse backgrounds participated. Feasibility targets were met: 77% recruitment of eligible women, 88% 12-week completion, and data completeness of 82% at 3 months and 79% at 6 months. Post-intervention diabetes distress decreased by 1.52 points (p < 0.001), self-reported HbA1c

**Data availability statement:** All relevant data are within the paper and its Supporting Information files.

**Funding:** This research was supported by the Turkish Ministry of National Education, The Republic of Turkiye for Postgraduate Study Abroad Program to support DB's PhD. https://yyegm.meb.gov.tr/ The seedcorn funding for IPN from King's College London funded recruitment participant and facilitators. https://www.kcl.ac.uk/nmpc The funders had no role in study design, data collection and analysis, decision to publish, or preparation of the manuscript. No grant numbers applicable to scholarship received by DB & IPN.

**Competing interests:** The authors have declared that no competing interests exist.

decreased by 13 mmol/mol (p < 0.02) and BMI by 0.6 kg/m$^2$ (p < 0.049). Menopausal symptoms declined by 13 points on the Greene scale (p < 0.001). Improvements were observed in diabetes self-efficacy (+7.4 points, p < 0.001), general health (+27.5 points, p < 0.01), sleep disturbance (−6.2 points, p < 0.002), and confidence in making dietary choices (+1.1 points, p < 0.02). Goal attainment scores indicated that 68% of participants achieved or exceeded their expectations.

## Conclusions

This feasibility study suggests that the WWDP+ is acceptable and feasible for midlife women with type 2 diabetes. The findings will directly inform the design, sample size, and recruitment strategies of a fully powered randomised controlled trial to evaluate WWDP+ effectiveness.

## Trial registration

ISRCTN: ISRCTN93338547. https://doi.org/10.1186/ISRCTN93338547

## Introduction

Globally the number of people living with diabetes is approximately 537 million adults, a number expected to rise to 643 million by 2030 [1]. Over 90% of people with diabetes have type 2 diabetes mellitus (T2DM) [1]. In the UK, concerning recent data show that 4.4 million people are living with T2DM [2]. Particularly, changes in diet and physical activity have led to its sharp increases [3]. Women face a higher burden of diabetes related risk factors. Throughout their lives, women experience more hormonal and bodily changes due to reproductive factors, which increase their risk of developing diabetes compared to men [4]. Additionally, the impact of diabetes complications is often more severe in women, with higher mortality rates [5,6].

Midlife, which runs from the late reproductive stage (35–40 years) to late post-menopause (<65 years) presents various health challenges [7]. Declining oestrogen and oestradiol hormones increase body fat leading to gain weight and a larger waist circumference which increases women's susceptibility to T2DM [8,9]. Hormonal changes can begin during the peri-menopausal period, several years before menopause, and may therefore affect women in their late 30s and 40s. As women transition to menopause, the associated hormonal changes may also have cardiovascular effects, further influencing diabetes-related risks [10]. Poor psychological functioning is also more common among midlife women who have multiple chronic illnesses and a higher body mass index [11]. The emotional burden which accompanies diabetes, such as distress related to diabetes management and progression [12] has also been shown to have a negative impact on their health-related quality of life [13] and act as a specific barrier to optimal self-care in women [14].

Given this, there is a clear need for interventions which are tailored to improve the health of midlife women living with T2DM. Innovative self-management interventions have been shown to improve diabetes and reduce associated complications [15,16].

For example, reducing people's diabetes distress through interventions can lead to better self-care and improvements in health [12]. One such intervention is the multi-modal Women's Wellness with Diabetes Programme (WWDP), designed to be delivered through an eHealth website and booklets, which targets the health needs of midlife women living with T2DM [17,18]. An international study of the WWDP in Australia and the UK evaluated the feasibility of participant recruitment and retention rates for the programme and confirmed initial efficacy in improving wellbeing outcomes [17,18]. However, it also demonstrated challenges with intervention content, sub-optimal participant engagement with the website and goal setting, and women's preference for peer support. In addition, healthcare professionals supported the idea of online delivery to address implementation issues.

After refining the WWDP in line with these recommendations and Patient and Public involvement (PPI) two novel components were added [19]: (1) individualised goal setting and (2) online peer support. The WWDP+ migrated to a web-based platform for greater interactivity alongside frequent support/reminder communications. In this study, feasibility referred to the practicality of delivering the WWDP+ and was operationalised as: (1) recruitment of the target sample within the planned timeframe, (2) retention of participants at 12 weeks, (3) attendance at the online peer group, and (4) completion of follow-up questionnaires. This study aimed to evaluate the feasibility of the newly developed WWDP+ for midlife women in the UK. Feasibility of the delivery of, and engagement with, the WWDP+ was one element. Feasibility of the research protocol for evaluating the WWDP+ was a second element. Lastly the study aimed to determine the effect signal of the programme for improving a range of outcomes which improved during the previous feasibility trial [17,18]. This aimed to inform a future RCT.

## Materials and methods

### Design

A nonrandomised single-arm design with a 12-week follow -up, all participants received the WWDP+ intervention. Feasibility outcomes were assessed at 3-months and 6-months post baseline while effective signal outcomes were assessed at baseline, 3-months and 6-months post baseline. The study protocol can be found in S1 File.

### Study setting and recruitment

Participants were recruited through a four week, funded, social media campaign targeting midlife women with T2DM through Facebook and Google adverts. Women were invited to respond if they were interested in engaging with diabetes research in any of the following ways 1) Advising our team on research directions (public and patient involvement), 2) participating in research or 3) becoming a peer supporter for other women living with T2DM. A Microsoft form was accessible for women with T2DM to complete, to provide their demographic and contact details and to indicate which activities they were interested in. Those interested in research participation were emailed and given a project weblink for further information. A study-specific webpage included the aim of the study, eligibility criteria, a participant information sheet, and a link for a screening questionnaire via Qualtrics [20].

Women confirming interest through the webpages were contacted, given the opportunity to ask further questions and then asked to confirm their eligibility within 24 hours by completing a brief screening question. The screening questions comprised basic data on their physical health, demographic information and the Diabetes Distress scale (DDS) [21] which had indicated a strong effect signal in the previous trial [17,18]. The DDS measures the extent of diabetes distress related to their medical regimen, emotional, physician care and interpersonal domains. Those scoring over 2 (from a scale of 1–5) on the DDS were eligible because the prior feasibility trial reduced elevated diabetes distress. Pre-specified threshold reflects the level at which distress becomes clinically meaningful, distinguishing participants experiencing minimal or no distress (<2) from those with moderate to high distress (≥2). Using this cut-off allows consistent classification of participants and aligns with established conventions for interpreting DDS scores [12]. Participants received the informed consent

form online and provided written consent by digitally signing it on the Qualtrics [20] within 7 days before participating in the study. All participant recruitment including follow up were conducted between May 2024 and December 2024.

Inclusion criteria were: 1) women living in the UK with access to diabetes care in the NHS, 2) aged ≥ 45 and ≤65 years with a current diagnosis of T2DM 3) taking insulin or oral medication, 4) receiving diabetes care only through their GP 5) able to read/speak English sufficiently to take part, 6) access to a computer or laptop, internet access and IT literacy (defined as being able to buy a product over the internet, search for a topic and send an email), 7) agreeing to having a Facebook account, 8) scored 2 or above on the Diabetes Distress scale [21]. Exclusion criteria were concurrent participation in another research study that could interfere with the assessment of the programme's outcomes.

### Sample size

The target sample size was based on guidelines for feasibility studies to test procedures and processes and to estimate some of the parameters for the definitive trial [22]. In line with methodological recommendations for pilot and feasibility trials [23,24] which indicate that relatively small samples can be adequate for estimating feasibility parameters, but larger samples are required when the primary aim is precise estimation of a standard deviation, this study aimed to recruit a total of 40 participants.

### Materials

Assessments were conducted at three time points including baseline (T1) and immediately post-intervention (T2) and 6-months post baseline (T3) of the primary outcomes, with goal setting assessments at 3- and 12-weeks through online survey via Qualtrics [20]. An online survey link was sent to each participant's email address. Completion of the baseline and post and follow up questionnaires required around 30–60 minutes. The programme materials were developed and delivered by the research team and peer supporters for peer group. Participants had access to the online peer support group between 9:00 am and 5:00 pm daily, with the peer group being moderated during these hours and programme resources available via the website. The 12-week intervention followed a structured weekly flow with different educational modules in the E-book and peer support sessions in the group (S2 File).

Outcomes which improved during the previous WWDP trial [17] were targeted. All baseline and post study DDS were assessed within 48 hours of receipt of the completed survey. If the score was 3.5 or over, indicating significant distress [12], participants were contacted by the research team and signposted to resources to support their psychological wellbeing, e.g., the Samaritans or GP depending on how immediate the research team perceived their need for support either over the phone or email.

### Feasibility outcome measures

Feasibility outcomes related to the new programme components, and therefore not evaluated in the previous trial [17], were considered primary outcomes:

- Recruitment rate via the social media campaign- calculated as the proportion of eligible women who consented to participate and completed baseline assessments.

- Intervention/Peer Group attendance – calculated as the proportion of participants who logged into the online peer group site on at least three occasions over the 12 weeks.

- Data quality – calculated as the proportion of participants who provided data at all three time points.

- Pre-defined progression criteria for the trial to assess feasibility were set at achieving at least a 50% recruitment rate, 70% program completion rate and data completeness of 70%.

**Secondary outcome measures (self-reported)**

- Diabetes Distress was measured using the Diabetes Distress Scale [21] that contains 17-items categorised into four subscales labelled as Emotional Burden, Regimen Distress, Interpersonal Distress and Physician Distress.

- Confidence in self-management was measured using the Diabetes Management Self-Efficacy Scale (DMSES UK) [25].

- The Short-Form health survey (SF-36) is 36 item self-report scale assessed health-related QOL across 8 domains: physical functioning, physical health role limitations, emotional problems, role limitations, pain, mental health, social functioning, general health [26].

- Assessing positive or negative lifestyle factors was measured using International Physical Activity Questionnaire (IPAQ) [27] assessed physical activity whereas measuring dietary behaviours on diabetes management were assessed through UK version Diabetes and Diet Questionnaire (UKDDQ) [28].

- To assess smoking prevalence, a subset of key questions from the Global Adult Tobacco Survey [29] were used.

- Participants' sleep activity and quality was measured using the 21-item General Sleep Disturbance Scale [30].

- Furthermore, the standard Greene Climacteric Scale assessed menopausal symptoms, with validated 21 items assessing subscales for vasomotor, somatic, psychological (anxiety and depression) with strong reliability [31].

- The Goal Assessment Scale (GAS) [32] was used to assess whether participants had achieved their goals within seven days of receiving the form in week 12 using a $+2$ to $-2$ scale.

- Physiological measures: Anthropometric measures covered body weight, height, waist and hip circumferences and body mass index (BMI). Women were provided with a video from reliable source by Diabetes UK [33] which demonstrated how to take the various measurements using a tape measure and scales, and how to calculate their BMI.

**The Women's Wellness in Diabetes Programme + (WWDP+) Intervention**

The WWDP+ [19] comprised three components a) website and e-book of 12 chapters/weeks including reflections, information and interactive exercises b) Personalised goal prioritisation activity and c) a moderated peer support Facebook group.

**Website and eBook component of WWDP+**

The WWDP+ is a 12-week multi-modal intervention delivered through an eHealth website [17–19]. The online intervention encourages women to reduce diabetes complications and promote healthy behaviours and is theoretically underpinned by Social Cognitive Theory [34] with an emphasis on behaviour change through mastery experiences, vicarious experiences, and verbal encouragement through self-efficacy theory. It includes a web interface, including podcasts, and an interactive electronic book (eBook) accessible from any electronic device offering intervention instructions and guides for participants, a logbook to record relevant health and lifestyle information weekly and factsheets on diabetes-specific selected topics. Table 1 outlines the targeted health knowledge and behaviours addressed in the intervention. Further information on the intervention can be found in S1 Table.

**Peer support component**

Online peer support is facilitated by membership of a closed Facebook group and provides the opportunity to meet other women who are living with T2DM, to share experiences and information, learn more about diabetes and living with a lifelong condition, and contribute to communal online activities. Participants were expected to log into the online peer group

**Table 1. Targeted health knowledge and behaviours [17].**

| Knowledge/ Behaviour | Recommendations |
|---|---|
| 1. Stress and psychological wellbeing | • Sleep<br>• Develop healthy stress management strategies<br>• Reduce anxiety and depression<br>• Manage Diabetes Distress |
| 2. Diabetes self-management | • Medication concordance<br>• Blood glucose management<br>• Managing clinical appointments |
| 3. Physical activity | • Be moderately physically active, equivalent to brisk walking for at least 30 minutes per day. As fitness improves, aim for 150–300 minutes of moderate intensity exercise per week<br> or<br>75 to 150 minutes of vigorous intensity physical activity per week<br>• Complete strength exercises on two or more days a week that work all the major muscles. |
| 4. Diet | • Consume at least 5 x 80g portions of a combination of fruit and vegetables per day. (UK National Health Service (NHS), 2018).<br>• Eat mostly foods of plant-based origin<br>• Limit consumption of energy-dense foods<br>• Avoid sugary drinks and snacks<br>• Limit intake of red meat<br>• Manage portion size of meals<br>• Consumption of recommended alcohol intake (NHS, 2017) |
| 5. Body fatness | • Be as lean as possible within the normal weight range<br>• Avoid weight gain and increases in waist circumference |
| 6. Smoking | • Smoking cessation |
| 7. Menopausal symptoms | • Management of menopausal symptoms |
| 8.Preventative health and risk screening | • Heart disease, eye health, renal health, breast, and gynae-cological health |

on at least three times and/or visit to the website once per week for four of the twelve weeks. Specialised trained peer supporters moderated the peer activity daily from Monday to Friday. These moderators were midlife women with T2DM who had undergone over ten hours of training, delivered by DB and an experienced peer support trainer. The training covered both peer support content and technical aspects of online moderation supported by a peer supporter guidance handbook.

## Personal goal setting component

As part of the intervention, participants received via email a "Personal Goal Evaluation" (PGF) form with a unique identification number and including a Goal Assessment Scale (GAS) [32]. Participants were encouraged to read week 1 and 2 of the eBook and collaborate with their peer group to set up their goals. They had until the end of week 3 to finalise their goals of participation. Participants were encouraged to set "SMART" (i.e., specific, measurable, attainable, realistic, and timely) goals and to focus preferably on one or two goals throughout the intervention. To enhance commitment, participants were also encouraged to think about potential barriers and solutions to their action-plans. They rated their confidence (from 1 to 10) in achieving their chosen goal. Those with a confidence level of less than 7 were encouraged to re-visit their goal or explore ways to boost their confidence. By week 3, participants were expected to be ready for goal implementation. Once participants selected personalised goals at 3 weeks, which were subsequently rated post-intervention using GAS. Each goal was scored on a five-point scale from –2 (much less than expected outcome) to +2

(much more than expected outcome), with 0 representing the expected level of achievement. For participants with multiple goals, scores were grouped according to common goals and analysed separately for each goal. Goals common to multiple participants were aggregated to show the overall goal attainment. This approach allowed goal attainment to be summarised both at the individual and goal-specific level.

## Data analysis

Descriptive statistics were used to characterize rates of demographic characteristics, recruitment, and data completeness. For continuous data, means and standard deviations were calculated, as well as frequency and percentages for categorical variables. To explore the potential impact of the intervention, statistical analyses were performed to assess changes in secondary outcomes over time. For normally distributed continuous variables through Shapiro-Wilk test, paired t-tests for variable on IPAQ and UKDDQ were used to compare pre- and post-intervention means, while the Wilcoxon signed-rank test for variables on DDS, DMSES UK, SF-36, HbA1c, BMI, weight, GSDS was applied for non-normally distributed data. Mean changes and standard deviations were reported, and where relevant, effect sizes were calculated to estimate the magnitude of change. Given that this study was not powered for hypothesis testing, statistical significance was not the primary focus; instead, results were interpreted with reference to Minimal Clinically Important Differences (MCID) from existing literature where available. Data were analysed using the Statistical Package for Social Sciences (SPSS) version 29.0 for Windows. Due to the nature of the intervention and the non-randomised design of the study, neither the participants nor the researchers delivering the intervention were blinded. However, the outcome assessor responsible for collecting follow-up data was blinded to participants' intervention exposure. Each participant was assigned a unique identification (ID) number, and all outcome data were stored and analysed using these blinded IDs. The outcome assessor accessed and evaluated results based solely on the de-identified dataset to minimise potential assessment bias. For each analysis, only participants with complete data at the relevant time points were included. For paired comparisons (pre-post-test and follow-up), participants were required to have data at both time points; those with missing data were excluded from that specific analysis (listwise deletion). There were no missing data within each time point, as all participants who were included completed all items at each assessment.

## Ethical approval

Ethical approval was granted by King's College London Ethics Committee (Ref: HR/DP-23/24–34435) on (05/03/2024) and (Ref: RESCM-23/24–34435) (10/05/2024). This trial was retrospectively registered at ISRCTN (Registration number: ISRCTN93338547). At the time of study initiation, the study was initially not considered to fall under the category requiring prospective registration. The authors confirm that all ongoing/completed and related trials for this intervention are registered.

## Results

### Baseline characteristics

Thirty-four midlife women with T2DM participated to the study. Participants had a mean age of 55.38 years±5.15. The mean duration of T2DM was 6.83±5.1 years and ranged from 1 to 20 years, with 65% of participants having a self-reported HbA1c level exceeding 7% (53 mmol/mol). Mean Body Mass Index (BMI) of the sample was 33.13±7.31.

Most participants had a family history of diabetes (n=25). Participants reported that 24/34 were treated with oral hypoglycaemic agents (OHA) only, 8/34 with OHA and insulin and 1/34 with insulin only. The majority of participants were married (n=20). Regarding educational attainment, 14/34 had completed undergraduate studies, 8/34 held postgraduate qualifications, and 8/34 had completed college education. In terms of working status, 13/34 women reporting working full-time and 12/34 women with working part-time. The detailed baseline demographic data is provided in Table 2.

**Table 2. Socio-demographic characteristic of participants (n = 34).**

| Variables | Mean | SD |
|---|---|---|
| **Age (years)** | 55.38 | 5.15 |
| **Sex assigned at birth** | n (Frequency) | % |
| Female | 34 | 100% |
| **Ethnicity n(%)** | | |
| White British | 20 | 58% |
| White other | 3 | 9% |
| Asian (Indian/Chinese) | 5 | 15% |
| Black Caribbean/African/British | 3 | 9% |
| Mixed | 3 | 9% |
| **Country of residence n(%)** | | |
| UK | 34 | 100% |
| **Nationality n (%)** | | |
| UK | 30 | 88% |
| Poland | 3 | 9% |
| United States | 1 | 3% |
| **Measurements** | Mean | SD |
| Weight (kg) | 91.09 | 22.61 |
| Height (cm) | 165.81 | 9.85 |
| Hip (cm) | 113.91 | 23.22 |
| Waist (cm) | 102.03 | 24.81 |
| Body Mass Index (BMI) | 33.13 | 7.31 |
| **Duration of diabetes** | Mean | SD |
| Years since diagnosis | 6.83 | 5.1 |
| **Family History of Diabetes n (%)** | | |
| Yes | 25 | 74% |
| No | 9 | 26% |
| **Marital status n (%)** | | |
| Single | 4 | 12% |
| Married | 20 | 58% |
| In a relationship | 6 | 18% |
| Divorced | 4 | 12% |
| **Education n (%)** | | |
| Secondary school | 2 | 6% |
| College degree/Diploma | 8 | 23% |
| University/Degree | 14 | 42% |
| Postgraduate degree | 8 | 23% |
| Other | 2 | 6% |
| **Employment status n (%)** | | |
| Employed full-time | 13 | 38% |
| Employed part-time | 12 | 35% |
| Home duties | 1 | 3% |
| Unemployed | 1 | 3% |
| Unable to work | 2 | 6% |
| Other | 5 | 15% |
| **Self-reported HbA1c n (%)** | | |
| Less than 6.5% (48 mmol/mol) | 6 | 18% |

*(Continued)*

**Table 2.** (Continued)

| Variables | Mean | SD |
|---|---|---|
| Between 6.5% to 7% (48 mmol/mol to 53 mmol/mol) | 5 | 15% |
| More than 7% (53 mmol/mol) | 22 | 64% |
| I don't know | 1 | 3% |
| **Self-reported Diabetes Treatment* n (%)** | | |
| Oral hypoglycaemic agents (OHA) | 24 | 73% |
| OHA and Insulin | 8 | 24% |
| Insulin only | 1 | 3% |
| **Diabetes Complications (self-report) n (%)** | | |
| Heart | 1 | 3% |
| Leg ulcers | 2 | 6% |
| Numbness or pain hand or legs | 5 | 15% |
| Eyesight | 5 | 15% |
| Kidney | 1 | 3% |
| **Other comorbid conditionsᵃ n (%)** | | |
| Menstrual problems | 2 | 6% |
| PCOS | 4 | 12% |
| Menopause | 6 | 18% |
| Anxiety | 13 | 38% |
| Depression | 7 | 21% |
| Other | 11 | 32% |
| **Diabetes education attended n (%)** | | |
| Yes | 15 | 44% |
| No | 19 | 56% |
| **Smoking currently n (%)** | | |
| Yes | 4 | 12% |
| No | 30 | 88% |

*Diabetes treatment was classified hierarchically, would be expected lifestyle advice including diet and exercise as part of usual care; therefore, the categories represent pharmacological treatment step-up.

ᵃFrequencies and percentages exceed the total sample size as some participants have multiple comorbidities

## Feasibility outcomes

Eighty-three women expressed interest in being a research participant and they were sent information about this study (Fig 1). Forty-three (52%) of these women were deemed ineligible through not meeting inclusion criteria or did not give their consent. The remaining forty were then invited to complete a baseline questionnaire. Six women did not complete the baseline assessment, and thirty-four women entered the study. The feasibility targets were assessed using pre-defined progression criteria; thus, the recruitment rate of 77% of eligible women (n=34/44) met the pre-determined recruitment target of 50% at baseline.

In terms of data completeness, 88% of participants completed the 3-week personal goal-setting forms, 82% completed the 12-week personal goal assessment forms, 82% completed the 3-month assessments, and 79% completed the 6-month post-baseline assessments. These completion rates exceeded the pre-determined threshold of 70%.

A total of 30 women (88%) completed the intervention. For peer group attendance rates, the data suggested that at least 28 women (82%) logged onto the online peer group on at least three occasions over the 12 weeks. This met the pre-determined peer group attendance rate of 70%. As Fig 2 suggests, despite stable group engagement during the first eight weeks of online peer group use, fluctuations were observed in the final four weeks during which fewer women

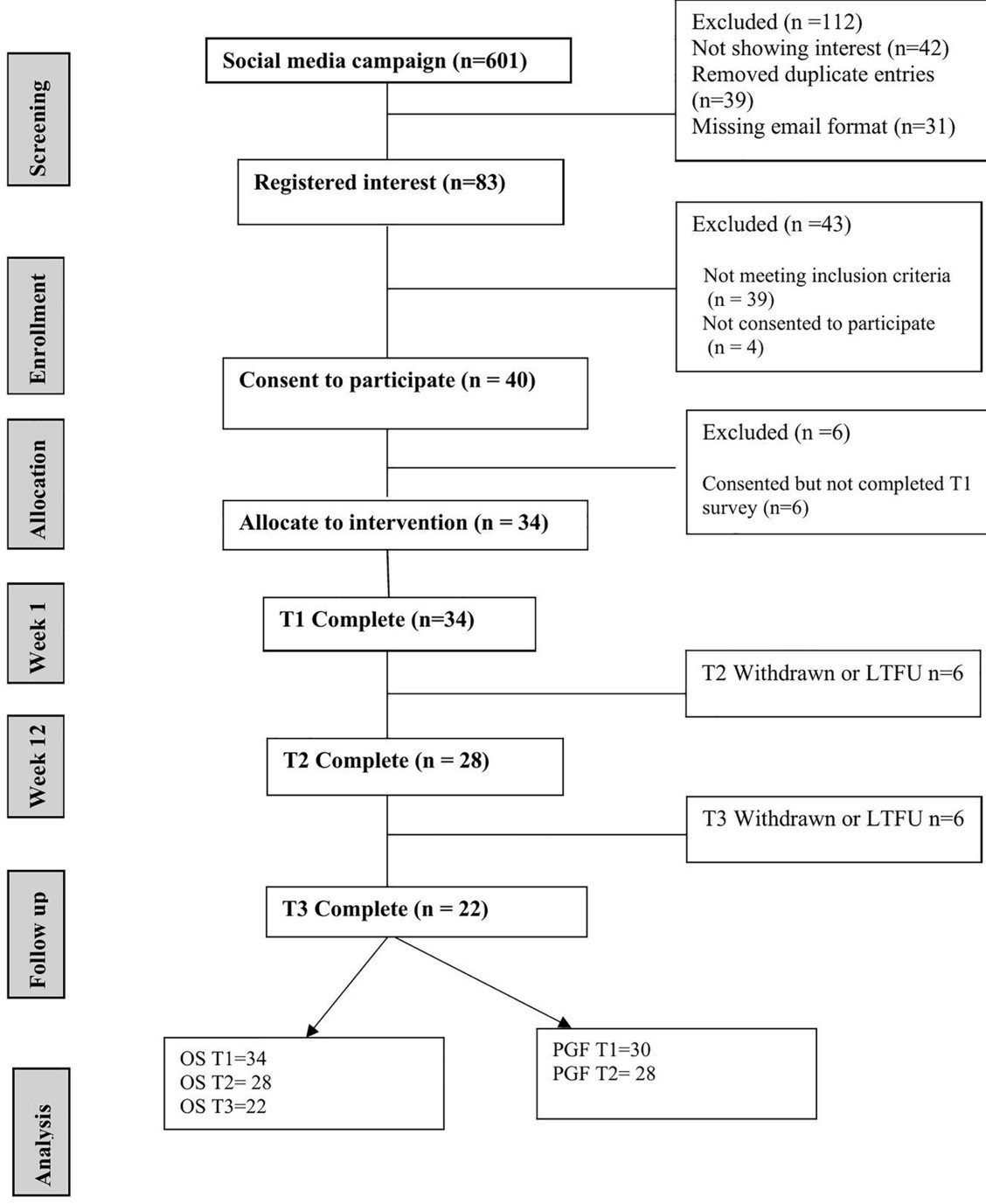

**Fig 1. CONSORT flow diagram.**

logged on. Of the 215 posts analysed, 43 were from peer supporters and were excluded from the analyses. The remaining posts, generated by participants, showed the highest engagement on Mondays, Tuesdays, and Wednesdays, mainly between 12:00 pm and 8:00 pm. This provides insight into peak activity periods. Further details of patterns of daily use can be found in Fig 2 and S1 Fig.

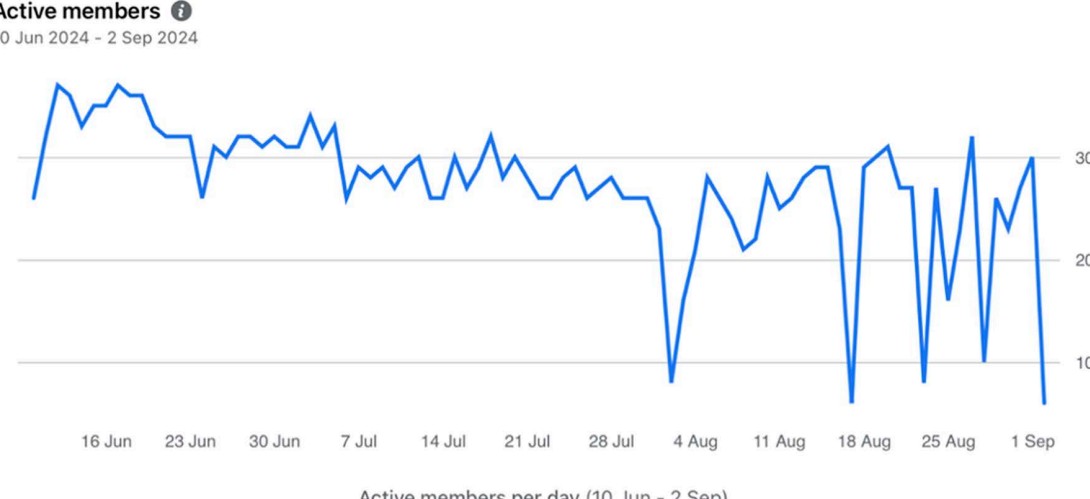

**Fig 2. Patterns of peer support over the 12-weeks intervention.**

## Goal setting findings

Among the 30 women that completed the goal setting by 3-week mark, 30% chose one goal while the majority (70%) selected two goals to focus on. Table 3 shows from the eight available options, participants chose the goals.

At T2, 28 women (82%) completed their personal goal assessment forms, with data showing that 61% had better than expected progress with their goal, 7% had made expected progress and 32% had fared worse than they had expected.

## Health related outcomes findings

Paired comparisons were conducted between T1 and T2, and between T1 and T3. Only participants with data available at both time points were included (Table 4). Participants experienced a mean reduction in diabetes distress by 1.52 from T1 to T3. Particularly, emotional burden (1.19), regimen distress (1.66), interpersonal distress (1.59), and physician distress (1.71) were found. These reductions exceeded the MCID of 0.25 established for the DDS [35]. Compared to the baseline,

**Table 3. The proportion of the preferred goals at week 3.**

| Preferred Goal | n = 30 | Frequency |
|---|---|---|
| Taking regular physical activity | 13 | 43% |
| Eating a healthy diet | 11 | 37% |
| Diabetes specific healthy behaviours<br>• Monitoring your own blood glucose (sugar) levels regularly (even if you cannot do this currently)<br>• Taking your diabetes medication<br>• Attending diabetes appointments for your eyes and foot checks and blood tests | 6 | 20% |
| Weight and Body mass Index | 5 | 17% |
| Reducing your Stress | 4 | 13% |
| Taking steps to help you sleep well | 4 | 13% |
| Reducing unhealthy habits | 4 | 13% |
| Managing menopause and its symptoms | 2 | 7% |

**Table 4. The health-related data of study participants across three time points of the study. Descriptive statistics are presented as mean±SD or median (IQR), as appropriate.**

| Measurement | Baseline(T1) n=34 Mean (SD) | Mdn [IQR] | 3 months post base-line(T2) n=28 Mean (SD) | Mdn [IQR] | 6 months post baseline(T3) n=22 Mean (SD) | Mdn [IQR] |
|---|---|---|---|---|---|---|
| Emotional Burden | 3.09 (1.00) | 3.00 [1.45] | 2.53(0.80) | 2.30[3.40] | 1.90[0.90] | 1.90[1.55] |
| Regimen Distress | 3.43 (0.96) | 3.50 [1.40] | 2.60(0.87) | 2.40[4.40] | 1.77[0.82] | 1.60[0.90] |
| Interpersonal Distress | 3.75 (1.00) | 3.66 [2.00] | 2.36(1.02) | 2.00[4.33] | 2.16[0.96] | 2.00[1.34] |
| Physician Distress | 3.41 (1.07) | 3.25 [1.81] | 2.52(0.91) | 2.50[4.00] | 1.70[0.76] | 1.75[1.44] |
| **Total DDS** | 3.38 (0.93) | 3.26 [1.43] | 2.52(0.75) | 2.32[2.82] | 1.86[0.71] | 1.73[1.38] |
| **DMSES UK** | 78.38 (29.92) | 78.00 [42.00] | 94.85(21.76) | 100.50[96.00] | 120.40[20.11] | 114.00[24.00] |
| Physical Functioning | 68.52(31.44) | 82.50[42.50] | 79.10(21.43) | 80.00[90.00] | 82.95[24.08] | 92.50[22.50] |
| Role-Physical | 52.94 (39.29) | 50.00[81.25] | 66.07(38.01) | 75.00[100.00] | 86.36[30.59] | 100.00[6.25] |
| Role-Emotional | 47.05 (42.73) | 50.00[100] | 70.23(38.85) | 100.00[100.00] | 84.86[32.09] | 100.00[8.25] |
| Energy-Fatigue | 35.29(23.51) | 37.50[40.00] | 48.57(22.96) | 50.00[85.00] | 54.54[21.43] | 55.00[31.25] |
| Emotional Well-being | 60.11(23.17) | 66.60[29.00] | 71.85(20.75) | 78.00[72.00] | 81.27[17.94] | 86.00[19.00] |
| Social Functioning | 59.19(30.35) | 62.50[40.63] | 70.53(27.04) | 75.00[100.00] | 88.63[19.25] | 100.00[25.00] |
| Pain | 55.95(27.52) | 57.50[45.63] | 68.21(23.85) | 67.50[77.50] | 82.04[24.40] | 90.00[23.13] |
| General Health | 42.05(24.21) | 42.50[43.75] | 52.50(26.99) | 50.00[95.00] | 69.54[29.75] | 75.00[55.00] |
| SF-36 | | | | | | |
| Psychological | 14.55 (7.14) | 13.00[9.00] | 8.96(5.79) | 8.00[23.00] | 5.95[5.97] | 9.50 [13.00] |
| Physical | 5.82(4.30) | 5.50 [7.00] | 3.67(2.77) | 3.50[11.00] | 2.59[2.73] | 4.50 [6.00] |
| Vasomotor | 4.00(2.59) | 4.00 [4.00] | 3.35(2.07) | 3.00[7.00] | 1.68[0.94] | 2.00 [2.50] |
| **Total GCS** | 24.38(11.35) | 24.50 [16.00] | 16.00(8.84) | 14.00[32.00] | 11.27[9.41] | 17.50[22.50] |
| Sleep onset latency | 3.23(2.55) | 2.50[5.25] | 2.60(2.54) | 2.00[7.00] | 6.90[7.17] | 10.50 [14.25] |
| Mid sleep awakening | 5.08(2.39) | 7.00[4.00] | 4.10(2.64) | 4.00[7.00] | 8.81[4.57] | 8.50[8.25] |
| Early awakening | 4.02(2.63) | 4.00[6.00] | 2.85(2.72) | 2.00[7.00] | 4.68[5.96] | 4.50[14.00] |
| Quality of sleep | 13.70(5.47) | 13.00[8.25] | 8.89(3.48) | 8.50[13.00] | 9.45[4.88] | 8.50[8.00] |
| Quantity of sleep | 6.16(3.03) | 7.00[3.25] | 7.14(4.94) | 6.00[20.00] | 4.63[4.05] | 4.00[6.75] |
| Use of sleep substances | 3.29(3.20) | 1.00[5.00] | 1.28(2.53) | 0.00[10.00] | 0.27[1.07] | 0.00[0.00] |
| Fatigue/alertness | 11.29(3.17) | 11.00[5.00] | 11.89(2.04) | 12.00[7.00] | 0.18[0.66] | 0.00[0.00] |
| **Total GSDS** | 46.20 (12.52) | 48.50[21.00] | 40.82(14.79) | 37.50[71.00] | 34.95[14.34] | 38.00[35.00] |
| **IPAQ (Physical activity in minutes per week)** | 2842.09(3556.47) | 1311.00[3687] | 3217(4303) | 1639.5[2769.75] | 2670.5[1575.4] | 2521.5[2141.25] |
| **UK DDQS** | **Mean (SD)** | **Mdn[IQR]** | **Mean (SD)** | **Mdn[IQR]** | **Mean (SD)** | **Mdn[IQR]** |
| Healthy dietary choices | 11.52(5.89) | 12.00[8.25] | 14.14(5.80) | 15.00[7.50] | 15.95[6.52] | 13.50[15.50] |
| Less healthy dietary choices | 5.11(3.46) | 5.00[5.25] | 3.46(2.68) | 4.00[4.00] | 4.68[3.84] | 4.00[7.25] |
| Unhealthy dietary choices | 3.38(3.60) | 2.00[2.25] | 2.03(1.79) | 2.00[3.00] | 1.40[2.03] | 1.50[3.75] |
| Importance to change diet | 8.07 (1.67) | 8.00[2.00] | 8.47(2.01) | 8.50[3.00] | 8.50[1.76] | 8.00[3.00] |
| Confidence level to change diet | 5.88 (2.60) | 6.00[4.00] | 7.21(2.74) | 7.50[4.00] | 8.22[2.38] | 8.50 [3.75] |
| Concerned about weight | **n=34** | **%** | **n=28** | **%** | **n=22** | **%** |
| Not concerned | 2 | 6 | 2 | 7 | 9 | 41 |
| Little concerned | 5 | 15 | 10 | 36 | 9 | 41 |
| Moderate concerned | 7 | 21 | 8 | 29 | 1 | 4 |
| Very concerned | 20 | 59 | 8 | 29 | 3 | 14 |

there was an increase in the mean of diabetes management self-efficacy as measured by the DMSES UK at T3 (42.02). Self-reported HbA1c levels also showed a mean decrease of 13 points exceeding the MCID for HbA1c of 4–5 mmol [36] indicating a clinically meaningful difference at T3. Furthermore, the MCID of health-related quality of life suggests three to five points [37,38], therefore, the findings on general health also exceed 27.49 points which showed significant improvement.

The Wilcoxon signed-rank and paired t-tests were run to compare the intervention effects between T1 and T3. Findings on DDS ($z = -3.81$; $p < 0.001$) suggested a clear trend towards improvements at T3. Subsequently, a change in sub-outcomes of DDS which are emotional burden ($z = -3.02$; $p < 0.003$), regimen distress ($z = -3.90$; $p < 0.001$), interpersonal distress ($z = -3.70$; $p < 0.001$), and physician distress ($z = -4.11$; $p < 0.001$) were observed at T3. For self-efficacy findings ($z = -3.86$; $p < 0.001$) indicated a difference between T1 and T3. A trend of improvement was observed in self-reported HbA1c levels ($z = -2.27$; $p < 0.02$). Moreover, self-reported quality of life ($z = 2.48$; $p < 0.01$) showed a potential change from T1 to T3. Findings also revealed reductions in menopausal symptoms ($z = -3.94$; $p < 0.001$).

Other assessments were observed in general sleep disturbance ($z = -3.10$; $p < 0.002$). Physical activity in minutes per week was a mean of 2670 (1575) and there were observations respectively on weight at T3 ($z = -1.77$; $p < 0.076$) and on BMI ($z = -1.96$; $p < 0.049$). In addition, the confidence levels of women about their ability to change diet were found mean of 8.22 ($p < 0.02$) indicating improvement.

Moreover, improvements were observed in their concerns regarding their weight. Indeed, most of the participants (86%) indicated either moderate, little or no concern, and only 14% of participants remained very concerned about weight at T3. The study also found a decreased mean in unhealthy dietary choices (2.43) and an increase in healthy dietary choices (4.43) post-intervention. 68% of participants indicated that they had achieved their individualised goal attainment score overall.

At T1, nearly 12% of participants stated they were current smokers and up the data indicated that 8% of participants were still smokers at T3. Only one participant indicated that they were seriously thinking about quitting smoking within a month after intervention. All scores are presented in detail in Table 4.

## Discussion

This study's primary aims of determining intervention and research protocol feasibility were established. The WWDP+ was well received by midlife women with T2DM in a web-based platform with goal setting and structured peer support. These feasibility findings align with those of the original programme [17]. Despite focusing only on women receiving primary care, high levels of co-morbidity were identified strengthening need and opportunities for supporting health and well-being in this population.

The WWDP+ programme included enhancements such as goal-setting support, peer group interactions, and tailored resources, which were not present in the previous version. These improvements likely enhanced participants' engagement and self-management by providing structured support, peer motivation, and personalised guidance, thereby increasing the programme's relevance and impact for midlife women with T2DM.

Goal setting was a central component with 70% of participants setting two personalised goals in which they exceeded their attainment expectations at T2. The intervention was effective in motivating sustained behavioural change aligning with existing evidence that structured goal setting, combined with social support, enhances diabetes self-management and leads to better health outcomes [39,40]. Despite 97% of participants taking diabetes medication, only 20% selected diabetes-specific healthy behaviour goals, suggesting a preference for broader wellbeing-framed goals alongside core self-management.

Goal setting with social support and action planning has been linked to reductions in BMI and improvements in physical activity levels among individuals newly diagnosed with T2DM [41]. Goal setting with access to the closed Facebook group enabled the discussion of goals with their peers which may have supported greater motivation.

The first eight weeks peer group engagement was stable with thirty interactions; these did reduce during the final month. Programme fatigue, participant motivation, or external influences may account for this [42].

Intervention effect signals revealed notable improvements from T1 to T3 in diabetes knowledge, self-management behaviours, diabetes distress and quality of life, BMI and weight, self-efficacy, sleep duration, menopausal symptoms and HbA1c. Moreover, diabetes distress and health-related quality of life met or exceeded the MCID.

Our baseline data showed that over 50% of our participant group had baseline very high levels of diabetes distress (scores of 3.5 or greater on the DDS [12]. Indeed, other studies have found that diabetes distress can be higher in women participants with T2DM than men [43,44]. The WWDP+ strengthened mastery and used vicarious experiences via the peer support group to reduce this distress. There is also an established link between fewer self-management behaviours and elevated diabetes distress, which can lead to increases in HbA1c levels [45]. Our participants reported above target HbA1c levels at T1 and a mean/median reduction of HbA1c at T3, meeting the MCID for this outcome. Studies confirm the importance of collecting HbA1c levels in any diabetes management interventions in order to provide insights into long-term glycemic control and informing clinical desicision making [46,47].

As a feasibility study, the current findings provide essential information for planning a future definitive trial. Data on recruitment and completion rates, as well as variability in outcome measures, will inform sample size calculations and study procedures. Additionally, patterns of goal selection and adherence provide insight into intervention acceptability and will guide refinement of the programme for a larger-scale study.

Potential confounding factors may have influenced the outcomes of this study. These include participants' baseline health status, prior experience with diabetes self-management, and external influences such as full-time and part-time employment, family responsibilities which may have affected their ability to engage fully with the programme.

Direct usage of the e-book was not systematically tracked. However, participant engagement was supported and partially monitored through multiple integrated components of the WWDP+. The lack of formal documentation for e-book usage is acknowledged as a limitation. Future studies should address this to better understand the relationship between e-book engagement and intervention outcomes. Our study faced also limitations in achieving meaningful outcomes in physical activity which has been found in other programmes [48] and required further exploration in this population.

Our social media campaign was largely successful in recruiting a diverse sample in terms of ethnicity and income levels. However, it may also prevent researchers from reaching target populations who do not use these kinds of online platforms. Similarly, using Facebook to provide the peer support group, may have prevented potential participants from joining the study. Finding other ways to recruit women who may also benefit from the revised WWDP but may not have an online presence on social media will be key in future work. Another limitation is that women who had previously participated in the WWDP were not specifically excluded from this study, and the data were not collected on prior of participation. Duration of diabetes (years since diagnosis) was collected at baseline as part of the demographic questionnaire but was not used as an inclusion criterion which may have influenced participants' responsiveness to the programme.

All the data collected in this study was self-report, which may introduce recall bias, social desirability bias, and response bias as limitation of the study. These factors could have led participants to under- or over-report their behaviours, potentially affecting the accuracy of the findings. In particular, the anthropometric measurements and clinical outcomes in this study were self-reported by women, and whilst we provided guidance as how to perform measurements, this may have compromised the reliability and validity of these findings [49]. For example, there may have been issues either with genuine errors or with women not feeling comfortable disclosing what they may have considered sensitive information such as weight. Although, most studies assessed the most frequently self-care behaviours, such as physical activity and diet [50] rather than measured self-care overall, as we did.

## Conclusion

This study suggests the feasibility and acceptability of WWDP+, as shown by promising recruitment and completion data. The findings indicate the potential of this adapted intervention which incorporates goal setting and peer support to

enhance a wide range of both public health and condition-specific outcomes. The collection of 6-month data informs the design of a fully powered trial to determine the effectiveness of this new intervention.

## Supporting information

**S1 Table. Participants read the Components from the Women's Wellness with Type 2 Diabetes Programme eBook.**
(DOCX)

**S1 Fig. WWDP+ Private Facebook Peer Group Engagement.**
(DOCX)

**S1 File. Study Protocol.**
(PDF)

**S2 File. TREND Checklist.**
(DOC)

## Acknowledgments

I would like to thank the peer support moderators. Thanks to the women who participated for their valuable time and data to evaluate the potential for the WWDP to improve the health for midlife women with type 2 diabetes. Thanks to Rebecca Rogers who shared the resources to design the peer supporter handbook. Thanks to Dr Samantha Coster for support in editing and proofreading this paper.

## Author contributions

**Conceptualization:** Deniz bozkurt, Maria Duaso, Jackie Sturt.

**Data curation:** Deniz bozkurt.

**Formal analysis:** Deniz bozkurt.

**Methodology:** Deniz bozkurt, Maria Duaso, Iliatha Papachristou Nadal, Jackie Sturt.

**Project administration:** Deniz bozkurt.

**Resources:** Rosie Walker.

**Supervision:** Maria Duaso, Iliatha Papachristou Nadal, Jackie Sturt.

**Visualization:** Deniz bozkurt.

**Writing – original draft:** Deniz bozkurt.

**Writing – review & editing:** Deniz bozkurt, Maria Duaso, Iliatha Papachristou Nadal, Rosie Walker, Jackie Sturt.

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
