## [Decision Letter · Decision Letter 0]

31 Jul 2025

PONE-D-25-24968Integrating Online Peer Support Group and Goal Setting to Promote Behavioural Changes in Midlife Women with Type 2 Diabetes: A feasibility study of the Women’s Wellness with Type 2 Diabetes ProgrammePLOS ONE

Dear Dr. bozkurt,

Thank you for submitting your manuscript to PLOS ONE. After careful consideration, we feel that it has merit but does not fully meet PLOS ONE’s publication criteria as it currently stands. Therefore, we invite you to submit a revised version of the manuscript that addresses the points raised during the review process.

Please address all comments provided by the reviewers (below and as per attached document) and carefully revise the manuscript, particularly the methodology section and presentation of the results to ensure that the manuscript is scientifically sound. It is also important for all typo and grammatically errors to be corrected at this point.

We look forward to receiving your revised manuscript.

Kind regards,

Shairyzah Ahmad Hisham, PhD.

Academic Editor

PLOS ONE

Journal Requirements:

2. We note that you have selected “Clinical Trial” as your article type. PLOS ONE requires that all clinical trials are registered in an appropriate registry (the WHO list of approved registries is at https://www.who.int/clinical-trials-registry-platform/network/primary-registries " https://www.who.int/clinical-trials-registry-platform/network/primary-registries and more information on trial registration is at http://www.icmje.org/about-icmje/faqs/clinical-trials-registration/ ). Please state the name of the registry and the registration number (e.g. ISRCTN or ClinicalTrials.gov ) in the submission data and on the title page of your manuscript. a) Please provide the complete date range for participant recruitment and follow-up in the methods section of your manuscript. b) If you have not yet registered your trial in an appropriate registry, we now require you to do so and will need confirmation of the trial registry number before we can pass your paper to the next stage of review. Please include in the Methods section of your paper your reasons for not registering this study before enrolment of participants started. Please confirm that all related trials are registered by stating: “The authors confirm that all ongoing and related trials for this drug/intervention are registered”. Please see http://journals.plos.org/plosone/s/submission-guidelines#loc-clinical-trials for our policies on clinical trials.

4. Please include your tables as part of your main manuscript and remove the individual files. Please note that supplementary tables (should remain/ be uploaded) as separate "supporting information" files”

5. Please remove all personal information, ensure that the data shared are in accordance with participant consent, and re-upload a fully anonymized data set.

Additional guidance on preparing raw data for publication can be found in our Data Policy (https://journals.plos.org/plosone/s/data-availability#loc-human-research-participant-data-and-other-sensitive-data) and in the following article: http://www.bmj.com/content/340/bmj.c181.long .

Reviewers' comments:

Reviewer's Responses to Questions

**Comments to the Author**

1. Is the manuscript technically sound, and do the data support the conclusions?

Reviewer #1: Yes

Reviewer #2: Partly

Reviewer #3: Yes

2. Has the statistical analysis been performed appropriately and rigorously? 

Reviewer #1: Yes

Reviewer #2: I Don't Know

Reviewer #3: Yes

3. Have the authors made all data underlying the findings in their manuscript fully available?

Reviewer #1: Yes

Reviewer #2: Yes

Reviewer #3: Yes

4. Is the manuscript presented in an intelligible fashion and written in standard English?

Reviewer #1: Yes

Reviewer #2: No

Reviewer #3: Yes

5. Review Comments to the Author

Reviewer #1: This research teams recruited 34 women aged between 45-65 with T2D for a single arm study to evaluate the feasibility of implementing online peer support group to promote behavioral changes. The results demonstrated its feasibility and acceptability.

1. Abstract. What’s WWDP? It would be better to attach your abbreviation when you spell them out for the first time before you use abbreviation in the text.

2. It might be informative to know whether there are other chronic diseases among the participants.

3. Line 160. Should be “from 1 to 10”

4. Paired t-test and Wilcoxon signed-rank test was implemented. It would be helpful to specify which variables were tested with which methods with normality assessment justification.

5. Please clarify how the missing data were handled in the analysis

6. Self-report data were used. It would be helpful to discuss potential biases and their impact on the findings.

7. A scoring method was implemented, e.g., GAS. It would be helpful to explain how these scores were aggregated or what they represent

8. Please clarify how pre-specified thresholds were determined.

9. As this is a feasibility study, it would be a great help to report or discuss how the resulting information serves as the resource for the future study, e.g., sample size planning.

Reviewer #2: Dear authors,

While I believe the study was well-conducted, I feel that the manuscript's reporting needs significant enhancement to improve clarity. Please refer to the attached document for my comments and suggestions.

Thank you.

Reviewer #3: Dear authors,

This study was conducted with the aim to evaluate the feasibility of a revised Women’s Wellness with Diabetes Program alongside peer support group and goal setting for midlife women to inform the design

of a future RCT. This is crucial in addressing issues concerning the T2DM status among midlife women in the near future. From the abstract, there was lack of explanation on the integration of peer support group into the program. There should also be a brief explanation on the goal setting for midlife women. For example "goal settings were made based on the personal goal evaluation form and the goal assessment scale (GAS). Participants were encouraged to set SMART short term goals throughout the intervention".

Overall, the introduction part was well-written with very good flow of issues discussed leading to the rationale of the study. There could be more details included under methods and discussion on the following:

1) Interventions provided: The information on delivery of the content was only obtained when referring to the supplementary document. It should be briefly described under methods i.e. how long does it take for the participants to answer the baseline questions, by whom were the materials delivered and prepared, the flow of delivery by day etc.

2) Statistical software or programs used - SPSS? Jamovi? not indicated in the manuscript

3) The interpretation of the discussion were not elaborated and sufficiently evidenced e.g. (Line 345) The sentence on limitations in achieving meaningful outcomes in physical activity was not explained and was left hanging. Important measures included in this clinical trial were BMI, HbA1c, glucose control. These measures are heavily associated with physical activity. An elaboration as to why it was not measured may provide justification on the design of the study.

Apart from that, line 329 stated that program fatigue, participant motivation or external influences may have reduced peer group engagement. Please include more explanation and evidences on this. The elaboration again will help to justify 'peer group engagement' as one of the intervention in improving T2DM among midlife women.

4) In supplementary document 2: Were the 215 posts came from participants only? It is understood that the facebook group is a private group. However, i am not sure whether the comments from the administrative personnel were included in the statistical analyses. There should also be some elaboration as to popular days and popular times for the comments to take place. This will help to strategize on when and how to approach participants appropriately.

5) There is also a concern on traceability of the engagement of participants with the T2DM programme e-book. Was this documented during the study period? If not, perhaps this could be explained as part of the study limitation. Documentation on this is definitely important as it will affect the study outcomes.

Although the intervention implemented was innovative and contextually relevant, the findings should be interpreted with caution. The study's reliance on subjective measurement and self-recorded measurement weakens the reliability of its conclusion. This was mentioned in the last paragraph of the discussion whereby it was stated that "all the data collected in the study-was self reported" which have compromised the reliability and validity of the findings. Improving study method by collecting anthropometric measurements and clinical outcomes during arranged follow-ups to ensure transparency and reliability of data may be suggested. Overall, the study was worth exploring.

6. PLOS authors have the option to publish the peer review history of their article (what does this mean? ). If published, this will include your full peer review and any attached files.

**Do you want your identity to be public for this peer review?** For information about this choice, including consent withdrawal, please see our Privacy Policy .

Reviewer #1: No

Reviewer #2: **Yes:** ZAINOL AKBAR ZAINAL

Reviewer #3: **Yes:** Aina Yazrin Ali Nasiruddin

---

## [Author Response · Author response to Decision Letter 1]

15 Sep 2025

Dear Reviewers;

We sincerely thank the reviewers for their constructive feedback. We have carefully revised the manuscript based on the suggestions provided.

Thanks in advance.

Deniz Bozkurt

---

## [Decision Letter · Decision Letter 1]

10 Dec 2025

PONE-D-25-24968R1The Women’s Wellness with Type 2 Diabetes Programme: Feasibility of an Online Peer Support and Goal-Setting Intervention for Midlife WomenPLOS One

Dear Dr. bozkurt,

Thank you for submitting your manuscript to PLOS ONE. After careful consideration, we feel that it has merit but does not fully meet PLOS ONE’s publication criteria as it currently stands. Therefore, we invite you to submit a revised version of the manuscript that addresses the points raised during the review process.

We look forward to receiving your revised manuscript.

Kind regards,

Aqeel M Alenazi

Academic Editor

PLOS One

Journal Requirements:

Additional Editor Comments:

Dear authors, please respond to the reviewers comments in order to make a decision for your manuscript

Reviewers' comments:

Reviewer's Responses to Questions

**Comments to the Author**

1. If the authors have adequately addressed your comments raised in a previous round of review and you feel that this manuscript is now acceptable for publication, you may indicate that here to bypass the “Comments to the Author” section, enter your conflict of interest statement in the “Confidential to Editor” section, and submit your "Accept" recommendation.

Reviewer #1: All comments have been addressed

Reviewer #2: (No Response)

Reviewer #3: (No Response)

2. Is the manuscript technically sound, and do the data support the conclusions?

Reviewer #1: (No Response)

Reviewer #2: Yes

Reviewer #3: Yes

3. Has the statistical analysis been performed appropriately and rigorously? 

Reviewer #1: (No Response)

Reviewer #2: Yes

Reviewer #3: Yes

4. Have the authors made all data underlying the findings in their manuscript fully available?

Reviewer #1: (No Response)

Reviewer #2: Yes

Reviewer #3: Yes

5. Is the manuscript presented in an intelligible fashion and written in standard English?

Reviewer #1: (No Response)

Reviewer #2: Yes

Reviewer #3: Yes

6. Review Comments to the Author

Reviewer #1: (No Response)

Reviewer #2: Dear author,

Thank you for considering my previous suggestions and comments for improvement of the manuscript. I have gone through the revised manuscript and I would like to confirm that most of the previous suggestions have been addressed accordingly. There are, however, a few more minor parts of the manuscript that need further revision. Please kindly refer to the attached document for your reference.

Reviewer #3: Dear Authors,

Overall the manuscript was well written. However, there are a few clarification required to further improve the manuscript.

Under the 'Materials and methods: Study Setting and recruitment' subheading, the duration of disease (from the point of diagnosis) was not included as part of the inclusion criteria. From my point of view, the duration of disease will greatly affect the outcomes of study. If it is included in the DDS, it would be great if this can be clearly outlined in the paragraph.

As for the sample size, could you please detail out the recommendations by reference no 23 and 24 in order to reduce the level of uncertainty in determining the sample size.

Referring to table 2, the ethnicity of the sample population was not based on universal categories e.g., chinese should be under Asian. White Turkish is also not a universal ethnic group classification. The classification under others can further be detailed out in methods.

In table 2, you may want to arrange the 'self-reported diabetes treatment' according to treatment step up therapy. Example, 1st group should be diet and exercise followed by OHA and then OHA and insulin and so on so forth. With this we can screen through the sample population according to severity of disease. I would also like to clarify, all diabetic patients should be on diet, exercise and lifestyle modification. Perhaps you could consider reclassifying the samples. However, i am aware that this will affect some parts of your discussion and data presentation.

Similarly in table 2, the reported diabetes complications are not the universal diabetes complications as per stated in the guidelines i.e. macrovascular and microvascular complications including diabetic retinopathy, nephropathy etc. The listed diabetes complications are more of comorbidities. Kindly revise.

Based on the above comments, there may be changes to the discussion part. Therefore i rest my further comments on the discussion part. All in all, this study offers beneficial outcomes in the management of diabetes among midlife women. Further studies should be conducted to cover a bigger sample population for better generalisability of the data.

7. PLOS authors have the option to publish the peer review history of their article (what does this mean? ). If published, this will include your full peer review and any attached files.

**Do you want your identity to be public for this peer review?** For information about this choice, including consent withdrawal, please see our Privacy Policy .

Reviewer #1: No

Reviewer #2: No

Reviewer #3: **Yes:** Aina Yazrin Ali Nasiruddin

---

## [Author Response · Author response to Decision Letter 2]

18 Dec 2025

We would like to thank the Editor and the reviewers for their constructive and helpful comments. We have carefully considered all suggestions and revised the manuscript accordingly. All changes are highlighted in the revised manuscript, and our point-by-point responses are provided in submission.

Best regards

Deniz Bozkurt

---

## [Decision Letter · Decision Letter 2]

20 Jan 2026

PONE-D-25-24968R2The Women’s Wellness with Type 2 Diabetes Programme: Feasibility of an Online Peer Support and Goal-Setting Intervention for Midlife WomenPLOS One

Dear Dr. bozkurt,

Thank you for submitting your manuscript to PLOS ONE. After careful consideration, we feel that it has merit but does not fully meet PLOS ONE’s publication criteria as it currently stands. Therefore, we invite you to submit a revised version of the manuscript that addresses the points raised during the review process.

**ACADEMIC EDITOR:** Please, revise according to reviewer's comments and double check on the percentages and numbers across the manuscript.

We look forward to receiving your revised manuscript.

Kind regards,

Aqeel M Alenazi

Academic Editor

PLOS One

Journal Requirements:

Additional Editor Comments:

Please, revise according to reviewer's comments and double check on the percentages and numbers across the manuscript.

Reviewers' comments:

Reviewer's Responses to Questions

**Comments to the Author**

1. If the authors have adequately addressed your comments raised in a previous round of review and you feel that this manuscript is now acceptable for publication, you may indicate that here to bypass the “Comments to the Author” section, enter your conflict of interest statement in the “Confidential to Editor” section, and submit your "Accept" recommendation.

Reviewer #2: All comments have been addressed

2. Is the manuscript technically sound, and do the data support the conclusions?

Reviewer #2: Yes

3. Has the statistical analysis been performed appropriately and rigorously? 

Reviewer #2: Yes

4. Have the authors made all data underlying the findings in their manuscript fully available?

Reviewer #2: Yes

5. Is the manuscript presented in an intelligible fashion and written in standard English?

Reviewer #2: Yes

6. Review Comments to the Author

Reviewer #2: Dear authors,

Thank you for addressing my previous suggestions. For this 2nd revision, I have suggested a few more minor amendments to be made in the Results section.

Thank you.

7. PLOS authors have the option to publish the peer review history of their article (what does this mean? ). If published, this will include your full peer review and any attached files.

**Do you want your identity to be public for this peer review?** For information about this choice, including consent withdrawal, please see our Privacy Policy .

Reviewer #2: No

---

## [Author Response · Author response to Decision Letter 3]

21 Jan 2026

Dear Reviewer,

Thank you for taking time to review my manuscript for third time. We have addressed all your remaining comments as seen in the attached table.

Best regards

Deniz Bozkurt

---

## [Decision Letter · Decision Letter 3]

9 Mar 2026

The Women’s Wellness with Type 2 Diabetes Programme: Feasibility of an Online Peer Support and Goal-Setting Intervention for Midlife Women

PONE-D-25-24968R3

Dear Dr. bozkurt,

We’re pleased to inform you that your manuscript has been judged scientifically suitable for publication and will be formally accepted for publication once it meets all outstanding technical requirements.

Kind regards,

Aqeel M Alenazi

Academic Editor

PLOS One

Additional Editor Comments (optional):

You addressed all the comments appropriately.

Reviewers' comments:

Reviewer's Responses to Questions

**Comments to the Author**

1. If the authors have adequately addressed your comments raised in a previous round of review and you feel that this manuscript is now acceptable for publication, you may indicate that here to bypass the “Comments to the Author” section, enter your conflict of interest statement in the “Confidential to Editor” section, and submit your "Accept" recommendation.

Reviewer #1: All comments have been addressed

2. Is the manuscript technically sound, and do the data support the conclusions?

Reviewer #1: (No Response)

3. Has the statistical analysis been performed appropriately and rigorously? 

Reviewer #1: (No Response)

4. Have the authors made all data underlying the findings in their manuscript fully available?

Reviewer #1: (No Response)

5. Is the manuscript presented in an intelligible fashion and written in standard English?

Reviewer #1: (No Response)

6. Review Comments to the Author

Reviewer #1: (No Response)

7. PLOS authors have the option to publish the peer review history of their article (what does this mean? ). If published, this will include your full peer review and any attached files.

**Do you want your identity to be public for this peer review?** For information about this choice, including consent withdrawal, please see our Privacy Policy .

Reviewer #1: No

---

## [Editor Report · Acceptance letter]

PONE-D-25-24968R3

PLOS One

Dear Dr. bozkurt,

I'm pleased to inform you that your manuscript has been deemed suitable for publication in PLOS One. Congratulations! Your manuscript is now being handed over to our production team.

Kind regards,

on behalf of

Dr. Aqeel M Alenazi

Academic Editor

PLOS One